# Differences in the proteomic profiles of the eutopic endometrium in patients with internal and external adenomyosis

Marta Valdés-Bango[1]*, Mikel Azkargorta[2], Meritxell Gracia[1], Cristina Ros[1], Mariona Rius[1], María-Angeles Martínez-Zamora[1], Cristian De Guirior[1], Lara Quintas[1], Félix Elortza[2], Francisco Carmona[1]

**1** Endometriosis Unit, Gynecology Department, Institut Clinic of Gynecology, Obstetrics and Neonatology (ICGON), Hospital Clínic of Barcelona, Universitat de Barcelona, Barcelona, Spain, **2** Center of Cooperative Research in Biosciences (CIC BioGUNE), Basque Research and Technology Alliance (BRTA), Bizkaia Science and Technology Park, Derio, Spain

\* martavaldesbc@gmail.com

## Abstract

Two separate phenotypes of adenomyosis have been recognized, determined by the anatomical position of the adenomyotic lesions within the myometrium. This suggests that adenomyosis impacting the inner myometrium and that affecting the outer myometrial layer may have distinct origins and display different clinical and radiological characteristics. We aimed to investigate the endometrial proteomic profiles of patients with both adenomyosis phenotypes to identify differentially expressed proteins and molecular pathways, shedding light on their distinct pathogenic mechanisms. We conducted a cross-sectional study that included thirty-six participants (nine with internal adenomyosis, nine with external adenomyosis, and eighteen healthy controls based on sonographic criteria) from September 2021 to September 2022. Endometrial samples were collected and processed for proteomic analysis. Mass spectrometry and a Data Independent Acquisition strategy were used to identify differentially expressed proteins. Gene Ontology and Ingenuity Pathway Analysis were employed for further functional analysis and network generation. The proteomic profiles of the eutopic endometrium differed significantly among women with internal adenomyosis, external adenomyosis, and controls. Biological functions related to the innate immune response were affected by differentially expressed proteins in patients with both phenotypes of adenomyosis compared to controls. The proteomic profiles of the endometrium of women with external versus internal adenomyosis exhibited significant differences, with external adenomyosis showing a heightened immune response and inflammatory activity, while internal adenomyosis was associated with altered signaling pathways related to cell migration and apoptosis. Upstream regulator analysis predicted the activation of inflammatory mediators like LPS, TGF-β1, IL-4, and IFN-γ in external adenomyosis, and MAPK1 and IRF2BP2 along with multiple

**Data availability statement:** The mass spectrometry proteomics data have been deposited to the ProteomeXchange Consortium via the PRIDE [1] partner repository with the dataset identifier PXD055131.

**Funding:** This study has been funded by Premio fin de residencia Emili Letang 2021 by Hospital Clínic de Barcelona received by M.V-B. (https://www.clinicbarcelona.org/en/docencia/residentes/post-residency-contracts). I confirm that the funder had no role in study design, data collection and analysis, decision to publish, or preparation of the manuscript.

**Competing interests:** The authors have declared that no competing interests exist.

microRNAs in internal adenomyosis. Overall, our findings support distinct pathogenic mechanisms for the two adenomyosis phenotypes, highlighting the need for further research to explore their implications for diagnosis, correlation with symptoms, and new potential therapeutic strategies.

## Introduction

Adenomyosis is a common benign gynecological condition causing abnormal uterine bleeding, dysmenorrhea, chronic pelvic pain, as well as infertility [1]. Histologically, adenomyosis is defined by the presence of ectopic endometrial glands and stroma surrounded by hypertrophic and hyperplastic myometrium [2]. Recent studies have defined two distinct phenotypes of adenomyosis based on the location of adenomyotic lesions within the myometrium: either predominantly in the inner layer (internal adenomyosis) or more confined to the outer layer (external adenomyosis) [3–7]. Patients with external and internal adenomyosis phenotypes present radiological and clinical-demographic differences [4,7]. Although the exact etiology of the disease remains debated, some authors suggest that internal adenomyosis results from endometrial invasion or *de novo* from metaplasia of embryonic or adult stem cell transformation within the myometrium, while external adenomyosis may occur through invasion from adjacent endometriotic lesions or from differentiation of endometrial stem cells deposited during retrograde menstruation [5,6,8–13].

Moreover, it is known that patients with adenomyosis present an immune dysregulation leading to chronic inflammation as well as imbalances in steroidal hormones [14]. Previous investigations have shown the existence of intrinsic abnormalities in the eutopic endometrium of patients with adenomyosis and neurogenesis, angiogenesis, fibrosis, cell proliferation and migration, impaired apoptosis and oxidative stress have been recognized as important contributing pathogenic factors [14–17]. In this context, we could hypothesize that the eutopic endometrium of patients with internal and external adenomyosis presents distinct abnormalities and dysregulated pathways.

In the past few years, omics analyses have emerged, offering insights into biological pathways involved in diverse medical conditions, while also aiding in the identification of potential diagnostic and prognostic biomarkers. In this context, various transcriptomic and proteomic studies have been performed to investigate potential mechanisms underlying the pathogenesis and pathophysiology of adenomyosis [18–31].

Considering this, the present study aimed to compare the endometrial proteomic profiles of patients with adenomyosis and controls and to identify differentially expressed proteins (DEPs) and molecular pathways in the eutopic endometrium from patients with internal and external adenomyosis phenotypes at proteomic level, providing new insights into the potentially differing pathogenic mechanisms of these two phenotypes of the disease.

## Materials and methods

### Ethics statement

Ethical approval was obtained from the Ethics Committee for Biomedical Research - Hospital Clínic de Barcelona (Approval Number: HCB/2021/0242). All the study participants provided written informed consent before the study started.

This observational cross-sectional study was performed in a tertiary referral center between 1st September 2021 and 1st September 2022. Patients were included in the study according to the inclusion and exclusion criteria specified below. Nine patients with sonographic diagnosis of internal adenomyosis and 9 with external adenomyosis were included in the study. Control group included 18 asymptomatic women who were visited for routine screening with confirmation for normal uterus and adnexa based on sonographic examination. The exclusion criteria were as follows: (1) postmenopausal status (2) previous diagnosis of autoimmune, inflammatory and/or neoplastic diseases, (3) use of hormone treatment over the last 3 months before sample collection (4) use of antibiotic or pre or probiotics in the 3 months prior to sample collection, (5) presence of myomas or polyps in the ultrasound examination, (6) current menstruation.

For the ultrasound evaluation, we used the same methodology described in our previously published works [7,32]. Adenomyosis was diagnosed according to the following criteria established by the Morphological Uterus Sonographic Assessment (MUSA) group [33,34]: myometrial cysts, myometrial hyperechoic islands and echogenic subendometrial lines and buds (direct criteria); and asymmetrical thickening of uterine walls, fan-shaped shadowing, translesional vascularity, irregular or interrupted junctional zone (JZ) and globular uterus (indirect criteria). Adenomyosis was diagnosed in the presence of two or more criteria, with at least one being direct. To distinguish between internal and external adenomyosis we employed the following sonographic categorization of the myometrium into three layers: the inner myometrium (corresponding to the JZ), the middle myometrium extending from the JZ to the venous and arterial arcuate vessels of the uterus, and the outer myometrium located between the arcuate vessels and the uterine serosa. The vascular arcuate, observed in the sagittal plane in two-dimensional transvaginal ultrasound (2D TVUS) with the application of color or power Doppler, served as a reference for the differentiation of uterine layers [35]. Haga clic o pulse aquí para escribir texto. Internal adenomyosis was defined when the JZ (inner myometrium) was affected with or without involvement of the middle myometrium and without affecting the outer myometrium (subserosal layer). External adenomyosis, on the other hand, was considered when it involved the outer myometrium, with or without involvement of the middle myometrium, without altering the JZ. The presence of ovarian and deep endometriosis (DE) was also assessed according to the International Deep Endometriosis Analysis (IDEA) group consensus [36]. For each patient, the following socio-demographic and clinical data were collected: age, ethnicity, body mass index (BMI), infertility, nulliparity. Dysmenorrhea and non-cyclic chronic pelvic pain (NCCPP) were also assessed using a numerical rating scale (NRS), where 0 represented no pain and 10 indicated unbearable pain. Additionally, the presence of heavy menstrual bleeding (HMB) was evaluated based on subjective patient assessments [37].

### Sample collection and processing

Women were scheduled for sample collection between days 3 and 10 of their menstrual cycle, with the proliferative phase confirmed by ultrasound. For sample collection, a sterile speculum was used, and the vagina and external cervix were swabbed with chlorhexidine. Endometrial samples were then collected using a double sheathed, sterile endometrial suction curette that was passed through the cervix to collect an endometrial biopsy, taking care to avoid contact with the vaginal wall and cervix. Biopsies were submerged in 1.0 mL of NaCl (9 mg/mL). Samples were transported to the laboratory on ice and processed within one hour. Endometrial samples were centrifuged at 3000 g at 4 °C for 10 minutes, and pellets were stored at −80 °C until analysis. Pellets were stored for up to six months, from collection and initial processing, until all participant samples were collected, and final analysis conducted.

### In-solution tryptic digestion

Protein was extracted by incubating the sample in a buffer containing 7M urea, 2M thiourea, and 4% CHAPS. Samples were incubated in this buffer for 30 min at room temperature under agitation and digested following the filter aided sample preparation (FASP) protocol described by Wisniewski *et al.* [38] with minor modifications. Protein was quantified using the Bio-Rad Protein Assay Dye Reagent Concentrate colorimetric assay (Bio-Rad), following manufacturer's instructions, and approximately 20 μg per sample were processed. Trypsin was added in 50mM ammonium bicarbonate to a trypsin:protein ratio of 1:10, and the mixture was incubated for overnight at 37$^o$C. Peptides were dried out in an RVC2 25 speedvac concentrator (Christ) and resuspended in 0.1% FA. Peptides were desalted and resuspended in 0.1% FA using C18 stage tips (Millipore, St. Louis, MO, United States) with a maximum peptide load of 2 μg prior to acquisition.

### Mass spectrometry analysis

Samples were analyzed in a timsTOF Pro with PASEF (Bruker Daltonics, Bremen, Germany) coupled online to a Evosep ONE liquid chromatograph (Evosep, Odense, Denmark). Approximately 200ng were directly loaded onto the Evosep ONE and resolved using the 60 samples-per-day protocol.

Data Independent Acquisition (DIA) data was processed with DIA-NN software v.1.8.1 [39] for protein identification and quantification using default parameters. Searches were carried out against a database consisting of *Homo sapiens* protein entries from Uniprot in library-free mode. Carbamidomethylation of cysteines was considered as fixed modification and oxidation of methionines as variable modification. Proteins identified with a FDR < 1% were considered for further analyses. Data was loaded onto Perseus platform [40] for data processing (log2 transformation, imputation) and statistical analysis (Two-tailed Student's t-test). Proteins with a p-value<0.05 were considered as DEPs for the corresponding comparison.

### Functional analysis and network generation

Gene Ontology analysis was used to identify possible molecular functions and to visualize the potential biological translation of DEPs. DAVID platform (https://david.ncifcrf.gov/) was used for this purpose.

Ingenuity pathway analysis (IPA) software (Qiagen, Redwood City, CA) was used for characterization of the molecular events lying behind the differential protein patterns under analysis. In this software, the calculated p-values determine the probability that the association between proteins in the dataset and a given process, pathway or upstream regulator is explained by chance alone, based on a Fisher's exact test (p-value <0.05 being considered significant). The activation z-score represents the bias in gene regulation that predicts whether the upstream regulator exists in an increased (positive values) or decreased (negative values) activation state, based on the knowledge of the relation between the effectors and their target molecule. Z-score values > 2 or <-2 were considered as significantly activated or inhibited, respectively.

### Statistical analysis

Principle component analysis (PCA), Heatmaps and Volcano plots were generated online using the SR Plot resource (https://www.bioinformatics.com.cn/) following the instructions provided in each section.

## Results

### Samples and participant characteristics

A total of 36 women were included in the study: 9 with internal adenomyosis, 9 in the external adenomyosis group, and 18 in the control group. The baseline socio-demographic and clinical characteristics of included women are shown in Table 1. Sonographic findings of included patients are presented in S1 Table.

**Table 1. Socio-demographic and clinical characteristics.**

| ID | Adeno-myosis | Adenomyosis phenotype | Cau-casian | Age | Body Mass Index | Nulli-parity | Dysmen-orrhea | Non chronic pelvic pain | Abnormal uter-ine bleeding | Infer-tility |
|---|---|---|---|---|---|---|---|---|---|---|
| p1 | No | – | Yes | 33 | 24.7 | Yes | 0 | 3 | No | No |
| p3 | No | – | Yes | 42 | 23.4 | No | 2 | 2 | No | No |
| p11 | No | – | Yes | 38 | 24.9 | No | 2 | 2 | No | No |
| p12 | No | – | Yes | 28 | 22.9 | Yes | 1 | 0 | No | No |
| p13 | No | – | No | 31 | 27.9 | No | 0 | 0 | No | No |
| p14 | No | – | Yes | 45 | 20.9 | No | 0 | 0 | No | No |
| p15 | No | – | No | 34 | 25.4 | No | 2 | 0 | No | No |
| p16 | No | – | Yes | 38 | 22.7 | Yes | 5 | 0 | No | No |
| p25 | No | – | No | 43 | 26.3 | Yes | 6 | 5 | No | No |
| p31 | No | – | Yes | 35 | 30.0 | No | 3 | 0 | No | No |
| p32 | No | – | Yes | 31 | 21.3 | No | 2 | 0 | No | No |
| p35 | No | – | Yes | 39 | 18.4 | No | 2 | 0 | No | No |
| p36 | No | – | Yes | 40 | 24.7 | No | 7 | 5 | No | No |
| p38 | No | – | Yes | 29 | 30.5 | No | 3 | 0 | No | No |
| p39 | No | – | Yes | 44 | 26.7 | No | 2 | 0 | No | No |
| p40 | No | – | Yes | 31 | 19.5 | Yes | 7 | 0 | No | No |
| p46 | No | – | Yes | 44 | 19.3 | No | 6 | 0 | No | No |
| p101 | No | – | Yes | 44 | 22.7 | Yes | 5 | 0 | No | No |
| p56 | Yes | Internal | Yes | 41 | 28.0 | Yes | 6 | 8 | Yes | No |
| p57 | Yes | Internal | Yes | 37 | 21.5 | No | 0 | 0 | No | Yes |
| p58 | Yes | Internal | Yes | 38 | 17.9 | Yes | 10 | 8 | Yes | Yes |
| p71 | Yes | Internal | Yes | 39 | 26.2 | Yes | 9 | 8 | No | Yes |
| p84 | Yes | Internal | Yes | 42 | 35.9 | Yes | 7 | 5 | No | No |
| p87 | Yes | Internal | No | 41 | 34.5 | Yes | 4 | 4 | Yes | No |
| p88 | Yes | Internal | No | 43 | 20.7 | No | 10 | 4 | No | No |
| p92 | Yes | Internal | Yes | 43 | 21.3 | Yes | 0 | 3 | Yes | No |
| p98 | Yes | Internal | Yes | 42 | 20.6 | Yes | 6 | 5 | Yes | Yes |
| p52 | Yes | External | Yes | 41 | 20.3 | Yes | 9 | 9 | Yes | No |
| p54 | Yes | External | Yes | 39 | 31.6 | No | 9 | 7 | Yes | No |
| p65 | Yes | External | Yes | 42 | 26.4 | Yes | 2 | 8 | No | No |
| p66 | Yes | External | Yes | 36 | 26.3 | No | 9 | 0 | Yes | Yes |
| p90 | Yes | External | Yes | 44 | 28.3 | Yes | 0 | 3 | No | No |
| p91 | Yes | External | No | 32 | 24.6 | Yes | 10 | 9 | Yes | Yes |
| p95 | Yes | External | Yes | 42 | 19.4 | Yes | 6 | 0 | No | No |
| p97 | Yes | External | Yes | 29 | 24.2 | No | 7 | 0 | Yes | No |
| p99 | Yes | External | Yes | 30 | 22.0 | No | 10 | 4 | Yes | No |

## Identification of differentially expressed proteins

We conducted a comparative proteomic analysis of eutopic endometrium among women with external adenomyosis, internal adenomyosis, and controls (S1 Data).

A total of 281 DEPs were identified between women with external adenomyosis and controls, and 91 DEPs were identified between women with internal adenomyosis and controls (Fig 1).

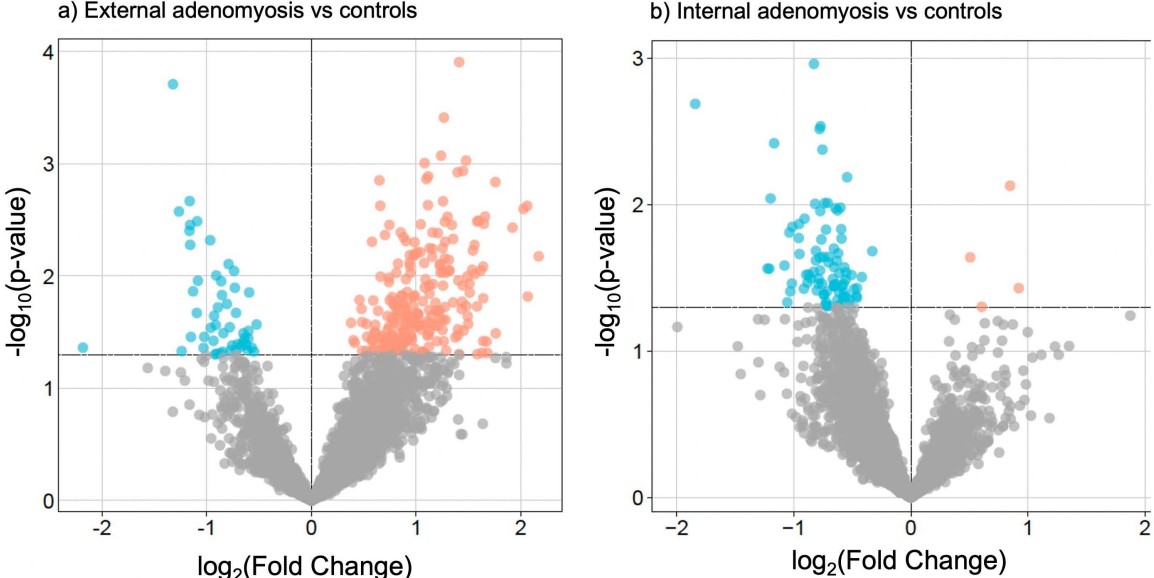

**Fig 1. Volcano-plots illustrating DEPs between (a) external adenomyosis and controls and (b) internal adenomyosis and controls.**

When focusing on adenomyosis phenotypes, 323 DEPs were identified between women with internal adenomyosis and those with external adenomyosis. Among these DEPs, 305 proteins were upregulated, and 18 proteins were down-regulated in the eutopic endometrium of patients with external adenomyosis compared to those with internal adenomyosis (Student's t-test, $p < 0.05$). The results reveal that eutopic endometrium from women with internal and external adenomyosis exhibit significantly different proteomic profiles (Fig 2).

**Functional analysis of differentially expressed proteins**

To further characterize the molecular events underlying different adenomyosis subtypes, a functional analysis of the DEPs detected in this work was performed.

First, NIH's DAVID tool was used for a broad determination of the biological and molecular processes these DEPs are mainly involved in. Consistent with the differences detected at the expression proteomics analysis, internal adenomyosis provided fewer enriched terms than external adenomyosis when compared to controls (S2 Data). In addition, these terms were appreciably different, except for GO:0045087 (Innate immune response), common to both internal and external adenomyosis.

DAVID analysis of the DEPs expressed in the comparison between internal vs external adenomyosis revealed an enrichment of terms related with complement activation, immunoglobulin production, immune response, zymogen activation and blood coagulation in external adenomyosis among others (Fig 3). In summary, these analyses revealed molecular and functional differences in the eutopic endometrium between internal and external adenomyosis.

Therefore, we decided to further characterize the molecular events underlying these differences using IPA software. This program uses both public and proprietary knowledge for a deeper and more precise characterization of the molecular events underlying the biological systems under analysis. In addition, the analysis of the protein expression fold change patterns adds an additional level of data interpretation through the Z-score. This parameter infers likely activation or inhibition states of the biological functions/pathways/upstream regulators based on the expression pattern afore mentioned.

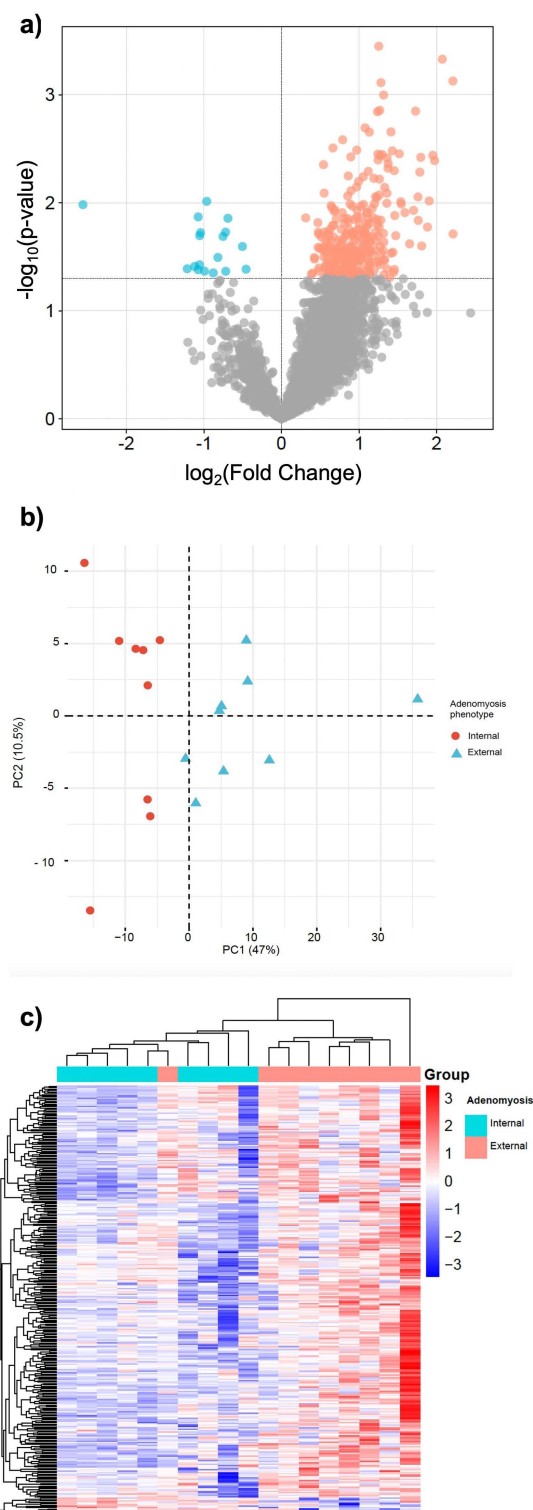

**Fig 2. DEPs between external adenomyosis and internal adenomyosis a) Volcano plot external vs internal b) PCA external vs internal c) Heatmaps external vs internal.**

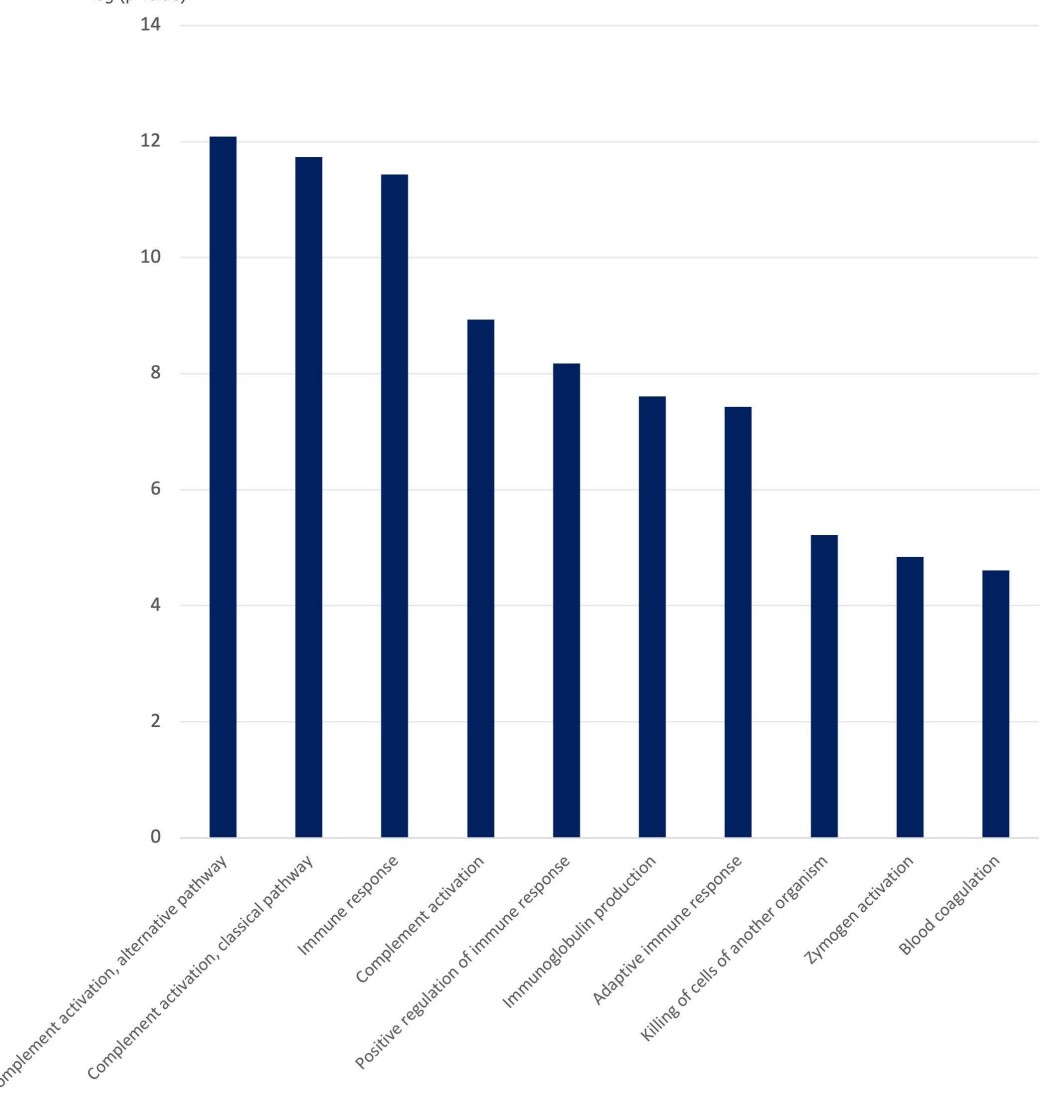

**Fig 3. Top 10 significantly enriched Gene Ontology Biological Processes (GO_BP) terms in external adenomyosis versus internal adenomyosis.**

The IPA results for the different analytical categories considered (Canonical Pathways and Upstream Regulators) are summarized in the S3 Data.

Fig 4 displays the most significantly upregulated and downregulated canonical pathways associated with DEPs in external versus internal adenomyosis. In external adenomyosis, key immune-related pathways were significantly activated, including the "Complement Cascade" (−log(p-value)=27.5; z-score = 3.6), "Fc gamma Receptor Dependent Phagocytosis" (−log(p-value)=22.9; z-score = 5.3), and "Neutrophil Degranulation" (−log(p-value)=22.7; z-score = 6.6); whereas pathways such as "Interleukin 12 (IL-12) Signaling and Production in Macrophages" (−log(p-value)=7.8; z-score = -3,2); "Programmed cell death protein 1 PD-1)/ Programmed cell death ligand 1 (PD-L1) Pathway" (−log(p- value)=4.1; z-score = -2.6); and " Rho GDP-dissociation inhibitor (RhoGDI) Signaling" (−log(p-value)=2.6; z-score = -2.4) were significantly inhibited.

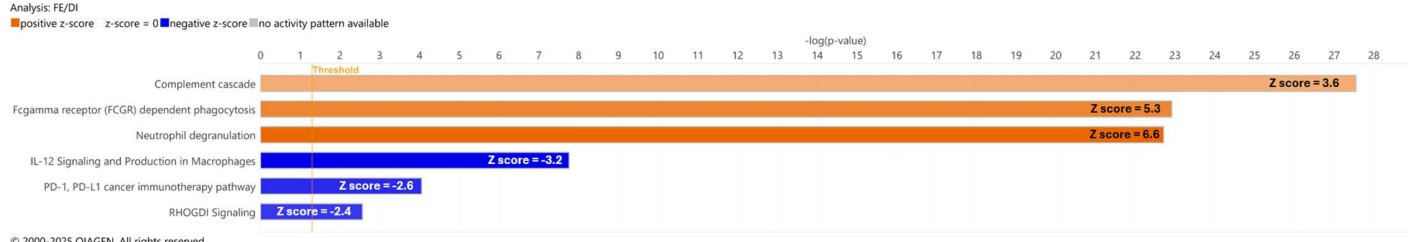

**Fig 4. Significantly activated (Z score>2) or inhibited (Z score<-2) canonical pathways associated with DEPs in external vs internal adenomyosis.**

Regarding significant differences among adenomyosis phenotypes in predicted upstream regulators (Tables 2 and 3), IPA identified notable activation of upstream regulators such as lipopolysaccharide (LPS) (p-value=2.39e-21, z-score=5.1), Transforming Growth Factor Beta 1 (TGFB1) (p-value=1.19e-10, z-score=2.7), Interleukin 4 (IL-4) (p-value=5.84e-10, z-score=4.0), and Interferon Gamma (IFGN) (p-value=1.15e-07, z-score=3.6), and inhibition of others such as Mitogen-Activated Protein Kinase 1 (MAPK1) (p-value=2.03e-02, z-score=-2.3), Interferon regulatory factor 2-binding protein 2 (IRF2BP2) (p-value=2.23e-02, z-score=-2.2), and multiple micro RNAs in external adenomyosis relative to internal adenomyosis.

## Discussion

The main findings of the present study are that: the proteomic profile of the eutopic endometrium differs significantly among women with internal adenomyosis, external adenomyosis, and controls; biological functions related to the innate immune response are affected by DEPs in patients with both phenotypes of adenomyosis compared to controls; the proteomic profiles of the endometrium of women with external versus internal adenomyosis also exhibit significant differences; multiple canonical pathways, upstream regulators and biological processes are differently affected in patients with external adenomyosis and those with internal adenomyosis.

The eutopic endometrium of women with adenomyosis (both internal and external) showed an enrichment of proteins related to innate immune response compared to controls. These findings align with data from Liu et al. (2008) [21] and Xiaoyu et al. (2013) [24], which identified abnormally expressed proteins associated with cell adhesion, apoptosis, and immune response in serum samples and adenomyotic lesions. Additionally, our results are consistent with Xiang et al. (2019) [18], who reported altered inflammatory and immune-related pathways ('IL-6 signaling,' 'ERK/MAPK signaling,' 'MIF regulation of innate immunity,' and 'LPS-stimulated MAPK signaling') in the eutopic endometrium of adenomyosis patients. More recently, Zhai et al. (2022) [27] identified dysfunction in the inflammatory response, extracellular matrix disassembly, and cell proliferation, along with dysregulated TNF, IL-17, and NF-κB signaling pathways in the endometrium of women with adenomyosis.

The eutopic endometrium from women with internal versus external adenomyosis shows significantly different proteomic profiles. We found an enrichment of proteins related with complement activation (by the alternative and classical pathways), immunoglobulin production and with blood coagulation in external adenomyosis. These results are not comparable with what has been reported in the literature because, to the best of our knowledge, this is the first publication comparing external and internal adenomyosis at a proteomic level.

Canonical pathway analysis is useful for understanding the biochemical and signal transduction pathways in which DEPs may participate. In external adenomyosis, key immune-related pathways were significantly activated, suggesting a heightened immune response, and increased inflammatory activity, particularly through complement activation, phagocytosis, and neutrophil involvement, when compared to internal adenomyosis. Previous studies have described an increased

**Table 2. Top 25 activated upstream regulators in external vs internal adenomyosis.**

| Upstream Regulator | Molecule Type | Predicted State | Activation z-score | p-value of overlap | Number of Target Molecules in Dataset |
|---|---|---|---|---|---|
| TGFB1 | Growth factor | Activated | 2,65 | 1,19E-10 | 58 |
| IL4 | Cytokine | Activated | 4,031 | 5,84E-10 | 45 |
| CEBPA | Transcription regulator | Activated | 2,123 | 1,69E-08 | 23 |
| IFNG | Cytokine | Activated | 3,569 | 1,15E-07 | 46 |
| IL2 | Cytokine | Activated | 2,188 | 3,03E-07 | 29 |
| IL5 | Cytokine | Activated | 3,925 | 5,09E-07 | 16 |
| TNF | Cytokine | Activated | 3,892 | 8,58E-07 | 53 |
| CD38 | Enzyme | Activated | 2,939 | 9,52E-07 | 12 |
| HNF4A | Transcription regulator | Activated | 2,308 | 1,24E-06 | 53 |
| IL1B | Cytokine | Activated | 2,229 | 4,27E-06 | 34 |
| BCR (complex) | Complex | Activated | 3,45 | 6,26E-06 | 15 |
| KLF6 | Transcription regulator | Activated | 2,545 | 6,95E-06 | 14 |
| NFKB1 | Transcription regulator | Activated | 2,779 | 8,79E-06 | 15 |
| CSF1 | Cytokine | Activated | 2,359 | 9,20E-06 | 17 |
| STAT1 | Transcription regulator | Activated | 3,607 | 2,50E-05 | 17 |
| SMARCA4 | Transcription regulator | Activated | 2,714 | 2,86E-05 | 22 |
| HNF1A | Transcription regulator | Activated | 2,537 | 3,08E-05 | 19 |
| CD40 | Transmembrane receptor | Activated | 3,024 | 4,42E-05 | 14 |
| RELA | Transcription regulator | Activated | 2,553 | 4,77E-05 | 18 |
| Brd4 | Kinase | Activated | 2,795 | 5,54E-05 | 8 |
| BHLHE40 | Transcription regulator | Activated | 3,208 | 6,11E-05 | 14 |
| KAT2B | Enzyme | Activated | 2,415 | 6,14E-05 | 6 |
| CCL20 | Cytokine | Activated | 2,236 | 1,42E-04 | 5 |
| CEBPD | Transcription regulator | Activated | 2,407 | 1,99E-04 | 8 |
| SREBF1 | Transcription regulator | Activated | 2,387 | 2,04E-04 | 12 |

abundance and activity of neutrophils in the eutopic proliferative endometrium of patients with endometriosis, as well as elevated levels of IL-8, which may enhance neutrophil recruitment [41]. Neutrophils have been associated with promoting angiogenesis through VEGF production and with immune modulation [41]. Furthermore, the complement system plays a crucial role in recognizing and clearing pathogens and apoptotic and necrotic cells, and in modulating the immune response [42].

On the other hand, in internal adenomyosis, IL-12 signaling may contribute to a persistent inflammatory microenvironment. Dysregulation of RhoGDI in internal adenomyosis, which regulates Rho GTPases involved in cell migration, adhesion, and proliferation and is related to multiple human cancers [43], could enhance the invasive properties of endometrial cells, contributing to their ability to penetrate and grow within the myometrium; interestingly, enhanced RhoGDI and PTEN have been recently described in adenomyosis and have been linked to pregnancy disorders in these patients [26]. Additionally, dysregulation of the PD-1/PD-L1 pathway, also observed in various cancers including endometrial cancer [44], might enable immune evasion, promoting the persistence and progression of the disease.

Furthermore, IPA results revealed differences in the predicted activity of upstream regulators when comparing the two phenotypes. Upstream regulators analysis helps to identify regulators that might be driving the observed changes in protein expression in our data set. In external adenomyosis, the analysis predicted the activation of multiple upstream regulators such as LPS, TGFB1, IL4, and IFNG. These mediators have been previously associated with adenomyosis [14] and suggest a heightened inflammatory response and fibrosis, in external adenomyosis. Interestingly, TGFB1, IL4, and

**Table 3. Top 25 inhibited upstream regulators in external vs internal adenomyosis.**

| Upstream Regulator | Molecule Type | Predicted State | Activation z-score | p-value of overlap | Number of Target Molecules in Dataset |
|---|---|---|---|---|---|
| SKIC2 | Enzyme | Inhibited | −2,828 | 5,72E-08 | 8 |
| ETV6-RUNX1 | Fusion gene/product | Inhibited | −2,103 | 1,31E-04 | 13 |
| OVOL2 | Transcription regulator | Inhibited | −2,222 | 3,27E-04 | 5 |
| CLPP | Peptidase | Inhibited | −2,433 | 3,77E-04 | 6 |
| IRGM | Enzyme | Inhibited | −2,425 | 8,03E-04 | 6 |
| HIBCH | Enzyme | Inhibited | −2,236 | 1,03E-03 | 5 |
| miR-155-5p (miRNAs w/ seed UAAUGCU) | Mature microrna | Inhibited | −2,21 | 1,29E-03 | 8 |
| NRAS | Enzyme | Inhibited | −2,382 | 1,46E-03 | 9 |
| FMR1 | Translation regulator | Inhibited | −2,178 | 2,29E-03 | 9 |
| TREX1 | Enzyme | Inhibited | −2,359 | 2,64E-03 | 7 |
| FOXA1 | Transcription regulator | Inhibited | −2,355 | 3,02E-03 | 9 |
| miR-16-5p (and other miRNAs w/seed AGCAGCA) | Mature microrna | Inhibited | −2,784 | 5,83E-03 | 8 |
| miR-291a-3p (and other miRNAs w/seed AAGUGCU) | Mature microrna | Inhibited | −2,204 | 6,86E-03 | 5 |
| mir-122 | Microrna | Inhibited | −2,236 | 9,09E-03 | 5 |
| let-7a-5p (and other miRNAs w/seed GAGGUAG) | Mature microrna | Inhibited | −2,646 | 1,21E-02 | 7 |
| PTPN11 | Phosphatase | Inhibited | −2,213 | 1,35E-02 | 6 |
| PD98059 | Chemical - kinase inhibitor | Inhibited | −2,727 | 1,38E-02 | 14 |
| mir-1 | Microrna | Inhibited | −2,219 | 1,59E-02 | 5 |
| MAPK1 | Kinase | Inhibited | −2,333 | 2,03E-02 | 10 |
| IRF2BP2 | Transcription regulator | Inhibited | −2,219 | 2,23E-02 | 6 |
| miR-124-3p (and other miRNAs w/seed AAGGCAC) | Mature microrna | Inhibited | −2,619 | 2,68E-02 | 7 |
| KDM5A | Transcription regulator | Inhibited | −2,449 | 3,13E-02 | 6 |
| mir-802 | Microrna | Inhibited | −2,236 | 3,32E-02 | 5 |
| RORC | Ligand-dependent nuclear receptor | Inhibited | −2,2 | 3,33E-02 | 6 |
| GABA | Chemical - endogenous mammalian | Inhibited | −3,162 | 3,61E-02 | 10 |

IFNG are cytokines also involved in the pathogenesis of DE through their effects on inflammation, tissue remodeling, fibrosis, cellular proliferation, and angiogenesis [45]. Conversely, the analysis predicted a notable activation of the upstream regulators MAPK1, IRF2BP2 and multiple microRNAs in internal adenomyosis compared to external adenomyosis. Xiang et al. had earlier reported the enrichment of the ERK/MAPK signaling pathway in adenomyosis [18]; additionally, the ERK/MAPK pathway has been linked to the proliferation of uterine smooth muscle cells in women with adenomyosis [46]. Recently studies published by Zipponi et al. [30] and Jia et al. [31] also suggest that the dysregulation of multiple

microRNAs can influence the progression of adenomyosis through the PI3K/AKT/mTOR pathway, which is involved in cell growth, survival, and proliferation. Moreover, recent data suggest IRF2BP2 involvement in the regulation of several cellular functions, such as the cell cycle, cell death, angiogenesis, inflammation, and immune response modulation [47]. Interestingly, microRNA let-7a-5p and microRNA-124-3p have previously been linked to adenomyosis [48] and chronic endometritis [49], respectively.

In recent years, it has been shown that patients with external and internal adenomyosis exhibit radiological and clinical differences. Although symptoms like bleeding and dysmenorrhea are common in both phenotypes, differences exist in other clinical parameters such as age, parity, infertility and the association with leiomyomas and DE [7,14,35,50]. In a previous study published by our team [7], we found that external adenomyosis is independently associated with the presence of DE and nulliparity, both of which support the theory of an ectopic endometrial origin, potentially exacerbated by retrograde menstruation and inflammatory processes. In contrast, internal adenomyosis, was strongly associated with leiomyomas. These findings, along with the results of our study showing DEPs and molecular pathways in the eutopic endometrium of patients with internal and external adenomyosis phenotypes at the proteomic level, support the hypothesis of distinct etiopathogenic mechanisms for both adenomyosis phenotypes.

Our study has several strengths that should be considered: to the best of our knowledge, it is the first study comparing endometrial proteomic profiles between external and internal adenomyosis phenotypes. Additionally, the study was performed in a tertiary referral center where the transvaginal ultrasound evaluations were performed by two expert sonographers with more than 10 years of experience and who have previously demonstrated a high diagnostic accuracy with transvaginal ultrasound for determining the presence of adenomyosis [51,52]. Moreover, all endometrial samples were collected by the same investigator following a strict protocol to avoid contamination. Our analytical approach leverages the timsTOF ion mobility mass spectrometry, enabling rapid and sensitive sample analysis. The use of DIA enhances sensitivity by addressing some of the limitations of the traditional Data-Dependent Acquisition (DDA) method. However, the approach may still be constrained by the sample's dynamic range, limited sample size, and analytical power, potentially leading to incomplete protein sampling. Functional analysis can help mitigate these limitations by highlighting key processes or regulators that may be crucial for understanding the observed differences. Another limitation of our study is that it would have also been interesting to include adenomyotic lesions and myometrial samples to better characterize both adenomyosis phenotypes at the proteomic level.

In conclusion, our results support the presence of abnormalities related to immune response, apoptosis, and cell migration in the eutopic endometrium of women with adenomyosis compared to controls. Furthermore, we demonstrated that the proteomic profile of the eutopic endometrium differs significantly between women with internal and external adenomyosis. Multiple canonical pathways, upstream regulators, and biological processes are differentially affected in these two adenomyosis phenotypes. In the eutopic endometrium of patients with external adenomyosis, a heightened immune response and increased inflammatory activity, particularly through complement activation, phagocytosis, and neutrophil involvement, seem to predominate. In the case of internal adenomyosis, other signaling pathways related to cell migration and apoptosis, along with immune dysregulation, appear to play a more significant role. Further studies are warranted to better understand the implications of these differences and their potential utility in diagnosing and treating adenomyosis.

## Supporting information

**S1 Table. Sonographic findings.**
(DOCX)

**S1 Data. Differential proteomics analysis results:** *Sample grouping*: **Correspondence between the samples analyzed and the groups considered in this work.** *Raw matrix*: original output from DIA-NN software. *Ctrl vs Internal, Ctrl*

*vs External and Internal vs External*: differential proteomics results for the corresponding comparisons. Column K contains the p-value for the Student's t-test and column L the corresponding expression ratio.
(XLSX)

**S2 Data.  Gene Ontology (GO) Analysis Results: DEPs were analyzed using the DAVID software tool.** Clustered results were generated, and the information was further processed manually. GO BP *Ctrl vs Internal, GO BP Ctrl vs External and GO BP Internal vs External*: significantly enriched (p<0.05) terms obtained from the original analysis for the corresponding comparisons.
(XLSX)

**S3 Data.  IPA analysis results: Summary of the Ingenuity Pathways Analysis (IPA) results for the internal vs external comparison.** *Can paths:* summary of significantly enriched canonical pathways, as defined by IPA. *Ups reg:* summary of the significantly enriched upstream regulators.
(XLSX)

## Author contributions

**Conceptualization:** Marta Valdés-Bango, Meritxell Gracia, Félix Elortza, Francisco Carmona.

**Data curation:** Marta Valdés-Bango, Mikel Azkargorta, Cristina Ros, Mariona Rius, María Angeles Martínez-Zamora, Cristian De Guirior, Lara Quintas.

**Formal analysis:** Marta Valdés-Bango, Mikel Azkargorta, Félix Elortza.

**Funding acquisition:** Marta Valdés-Bango, Meritxell Gracia.

**Investigation:** Marta Valdés-Bango, Meritxell Gracia, Cristina Ros, Mariona Rius, María Angeles Martínez-Zamora, Cristian De Guirior, Lara Quintas.

**Methodology:** Marta Valdés-Bango, Mikel Azkargorta, Meritxell Gracia, Francisco Carmona.

**Project administration:** Marta Valdés-Bango.

**Software:** Mikel Azkargorta.

**Supervision:** Francisco Carmona.

**Writing – original draft:** Marta Valdés-Bango, Mikel Azkargorta.

**Writing – review & editing:** Marta Valdés-Bango, Mikel Azkargorta, Meritxell Gracia, Cristina Ros, Mariona Rius, María Angeles Martínez-Zamora, Cristian De Guirior, Lara Quintas, Félix Elortza, Francisco Carmona.

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
