## [Decision Letter · Decision Letter 0]

PGPH-D-24-02496

DIFFERENCES IN THE PROTEOMIC PROFILES OF THE EUTOPIC ENDOMETRIUM IN PATIENTS WITH INTERNAL AND EXTERNAL ADENOMYOSIS

Dear Dr. Valdes-Bango Curell,

Thank you for submitting your manuscript to PLOS Global Public Health. After careful consideration, we feel that it has merit but does not fully meet PLOS Global Public Health’s publication criteria as it currently stands. Therefore, we invite you to submit a revised version of the manuscript that addresses the points raised during the review process.

Please note that we have only been able to secure a single reviewer to assess your manuscript. We are issuing a decision on your manuscript at this point to prevent further delays in the evaluation of your manuscript. Please be aware that the editor who handles your revised manuscript might find it necessary to invite additional reviewers to assess this work once the revised manuscript is submitted. However, we will aim to proceed on the basis of this single review if possible.

The reviewers have raised a number of concerns that need attention. Could you please revise the manuscript to carefully address the concerns raised?

We look forward to receiving your revised manuscript.

Kind regards,

Johanna Pruller, Ph.D.

PLOS Staff Editor

Journal Requirements:

1. We noticed you have some minor occurrence of overlapping text with the following previous publication(s), which needs to be addressed:

https://pubmed.ncbi.nlm.nih.gov/38661227/

In your revision ensure you cite all your sources (including your own works), and quote or rephrase any duplicated text outside the methods section. Further consideration is dependent on these concerns being addressed.

2. Please provide a/amend your detailed Financial Disclosure statement. This is published with the article. It must therefore be completed in full sentences and contain the exact wording you wish to be published.

**Please only choose the relevant sentences from below**

1. Please clarify all sources of funding (financial or material support) for your study. List the grants (with grant number) or organizations (with url) that supported your study, including funding received from your institution. 

2. State the initials, alongside each funding source, of each author to receive each grant.

3. State what role the funders took in the study. If the funders had no role in your study, please state: “The funders had no role in study design, data collection and analysis, decision to publish, or preparation of the manuscript.”

4. If any authors received a salary from any of your funders, please state which authors and which funders.

3. Your current Financial Disclosure states, “The study was funded by the scholarship “Emily Letang”—Hospital Clinic of Barcelona, granted in May 2021.”. However, your funding information on the submission form indicates is missing. Please indicate by return email the full and correct funding information for your study and confirm the order in which funding contributions should appear. Please be sure to indicate whether the funders played any role in the study design, data collection and analysis, decision to publish, or preparation of the manuscript.

4. Thank you for uploading your study's underlying data set. Unfortunately, the repository you have noted in your Data Availability statement does not qualify as an acceptable data repository according to PLOS's standards.

5. Please insert an Ethics Statement at the beginning of your Methods section, under a subheading 'Ethics Statement'. It must include:

1) The name(s) of the Institutional Review Board(s) or Ethics Committee(s)

2) The approval number(s), or a statement that approval was granted by the named board(s) 

3) (for human participants/donors) - A statement that formal consent was obtained (must state whether verbal/written) OR the reason consent was not obtained (e.g. anonymity). NOTE: If child participants, the statement must declare that formal consent was obtained from the parent/guardian.

6. Please provide an Author Summary. This should appear in your manuscript between the Abstract (if applicable) and the Introduction, and should be 150–200 words long. The aim should be to make your findings accessible to a wide audience that includes both scientists and non-scientists. Sample summaries can be found on our website under Submission Guidelines:

https://journals.plos.org/globalpublichealth/s/submission-guidelines#loc-parts-of-a-submission

7. We have noticed that you have a list of Supporting Information legends in your manuscript. However, the Supplementary File 1's label and caption is written as "Supplementary file 2. Sonographic findings". Please upload them as separate files with the item type 'Supporting Information'. 

Additional Editor Comments (if provided):

Reviewers' comments:

Reviewer's Responses to Questions

**Comments to the Author**

1. Does this manuscript meet PLOS Global Public Health’s publication criteria ? Is the manuscript technically sound, and do the data support the conclusions? The manuscript must describe methodologically and ethically rigorous research with conclusions that are appropriately drawn based on the data presented.

Reviewer #1: Yes

2. Has the statistical analysis been performed appropriately and rigorously?

Reviewer #1: Yes

3. Have the authors made all data underlying the findings in their manuscript fully available (please refer to the Data Availability Statement at the start of the manuscript PDF file)?

Reviewer #1: Yes

4. Is the manuscript presented in an intelligible fashion and written in standard English?

Reviewer #1: Yes

5. Review Comments to the Author

Reviewer #1: The study aims to investigate the proteomic profiles of different adenomyosis phenotypes, which is a well-defined and relevant objective given the clinical implications of adenomyosis. I have some comments listed below to be answered before the consideration of publication.

1. Regarding the sonographic findings among different groups, all patients in the internal In group have irregular or interrupted junctional zone, while all the patients in the external group are not. In general, the thickness of the junctional zone over 12 mm is in favor of adenomyosis, and abnormal junctional zone are thought to involve in the pathogenesis of adenomyosis. Have the authors considered to measure and compare the thickness of junctional zone in internal and external groups? Do authors think there will be a difference regarding the thickness of junctional zone between internal and external adenomyosis?

2. The authors mentioned that this is the first study comparing external and internal adenomyosis at a proteomic level and the findings support distinct pathogenic mechanisms for the two phenotypes. Based on the proteomic findings in the external adenomyosis, how will the authors explain the pathogenesis of external adenomyosis and correlate the findings with symptoms in external adenomyosis? What is the biggest difference regarding the pathogenic mechanism compared to internal adenomyosis?

Some minor suggestions:

3.Do authors miss internal in the title of Table 2 and 3?

- Table 2. Top 25 activated upstream regulators in external vs xx adenomyosis

- Table 3. Top 25 inhibited upstream regulators in external vs xx adenomyosis

4.In Figure 4, the threshold line is too thin to recognize, especially the figure 4a, considering the bar color is orange, the orange threshold line is too vague.

5.In Figure 3, it seems a bracket is missing in the Y axis. -log p value should be -log(p-value)

6.In supplementary file 1, there is typo about '' interrumped '' should be '' interrupted''

Overall, this study provides valuable insights into the distinct proteomic profile of internal and external adenomyosis, laying the groundwork for future research and potential diagnostic and therapeutic advancements .

6. PLOS authors have the option to publish the peer review history of their article (what does this mean? ). If published, this will include your full peer review and any attached files.

**Do you want your identity to be public for this peer review?** For information about this choice, including consent withdrawal, please see our Privacy Policy .

Reviewer #1: No

---

## [Decision Letter · Decision Letter 1]

PGPH-D-24-02496R1

DIFFERENCES IN THE PROTEOMIC PROFILES OF THE EUTOPIC ENDOMETRIUM IN PATIENTS WITH INTERNAL AND EXTERNAL ADENOMYOSIS

Dear Dr. Valdes-Bango Curell,

Thank you for submitting your manuscript to PLOS Global Public Health. After careful consideration, we feel that it has merit but does not fully meet PLOS Global Public Health’s publication criteria as it currently stands. Therefore, we invite you to submit a revised version of the manuscript that addresses the points raised during the review process.

As you will see in the appended and attached reports, your most recent manuscript has been assessed by the two previous reviewers, and both have raised further comments to address.

Furthermore, we note that one or more reviewers has recommended that you cite specific previously published works. As always, we recommend that you please review and evaluate the requested works to determine whether they are relevant and should be cited. It is not a requirement to cite these works. We appreciate your attention to this request.

We look forward to receiving your revised manuscript.

Kind regards,

Jason Morgan

Staff Editor

Journal Requirements:

Additional Editor Comments (if provided):

Reviewers' comments:

Reviewer's Responses to Questions

**Comments to the Author**

1. If the authors have adequately addressed your comments raised in a previous round of review and you feel that this manuscript is now acceptable for publication, you may indicate that here to bypass the “Comments to the Author” section, enter your conflict of interest statement in the “Confidential to Editor” section, and submit your "Accept" recommendation.

Reviewer #1: All comments have been addressed

Reviewer #2: (No Response)

2. Does this manuscript meet PLOS Global Public Health’s publication criteria ? Is the manuscript technically sound, and do the data support the conclusions? The manuscript must describe methodologically and ethically rigorous research with conclusions that are appropriately drawn based on the data presented.

Reviewer #1: Yes

Reviewer #2: Yes

3. Has the statistical analysis been performed appropriately and rigorously?

Reviewer #1: Yes

Reviewer #2: Yes

4. Have the authors made all data underlying the findings in their manuscript fully available (please refer to the Data Availability Statement at the start of the manuscript PDF file)?

Reviewer #1: Yes

Reviewer #2: Yes

5. Is the manuscript presented in an intelligible fashion and written in standard English?

Reviewer #1: Yes

Reviewer #2: Yes

6. Review Comments to the Author

Reviewer #1: This manuscript makes a contribution by pioneering the comparative proteomic analysis of endometrial tissues between internal and external adenomyosis phenotypes. The authors appropriately acknowledge limitations, including dynamic range constraints and sample size. Their suggestion to include adenomyotic lesions and myometrial samples in future research demonstrates forward thinking. This work establishes a foundation for targeted diagnostic and therapeutic approaches for adenomyosis subtypes, making it worthy of publication.

Some small corrections and points to reduce the redundancy of the text are listed below.

-Line 286 ‘dysfunction in the inflammatory response´, ‘the’ is missing.

-Line 289-291 ´ When focusing on the comparison between external and internal adenomyosis phenotypes, our results revealed that the eutopic endometrium from women with internal and external adenomyosis exhibits significantly different proteomic profiles´. The sentence is too redundant and could be simplified and shortened as the example, ´ The eutopic endometrium from women with internal versus external adenomyosis shows significantly different protein profiles´

-Line 336 ´common to both phenotypes’ the preposition should be ‘in’

Reviewer #2: - Please, carefully revised the references. Ref 15 and 48 are exactly the same paper. If the same first author and same year of publication, put a and b to differentiate them. Also, be consistent and report all references in the same way (using numbers). Don't use author's names like lines 71, 82, and 89.

- Line 78: add a reference to this sentence 'it is known that patients with adenomyosis present an immune dysregulation leading to chronic inflammation as well as imbalances in steroidal hormones'

- Line 89: Among transcriptomic studies, you may want to mention this paper that just came out:

Zipponi M. et al. Endometrial stromal cell signaling and microRNA exosome content in women with adenomyosis. Mol Hum Reprod. 2025 Jan 17;31(1):gaae044. doi: 10.1093/molehr/gaae044.

They found that miR-1 was downregulated in adenomyosis. This information could be implemented in lines 333-334 to add value and quality to your results reported in table 3.

I suggest to cite this paper n lines 327-328 when talking about ERK/MPAK enrichment in adenomyosis, as well as this very recent paper:

Jia, JJ., Lai, Hj., Sun, BW. et al. miR-21 regulates autophagy and apoptosis of ectopic endometrial stromal cells of adenomyosis via PI3K/ AKT/ mTOR pathway. Sci Rep 15, 7639 (2025). doi: https://doi.org/10.1038/s41598-025-92526-3.

- Line 151: for how long did you store fresh pellet at -80°C prior to analysis?

- Line 163: how did you quantify 200 ng? This information is missing. Please, revise this paragraph 'Mass spectrometry analysis' which is confusing. Did you quantify proteins before loading them into the liquid cromatograph or this type of information was retrieved after with DIA software? It is not clear at all.

- Line 185:

- Line 304: add a reference to this sentence 'previous studies have described an increased abundance and activity of neutrophils in the eutopic proliferative endometrium of patients with endometriosis, as well as elevated levels of IL-8, which may enhance neutrophil recruitment'

- Write the word 'p-value' in the same way in the manuscript and in the figures.

Figures:

Figure 1.

The volcano plots should have the same scale for the x-axis.

Figure 3.

Define GO_BP in the legend as gene ontology_biological processes and explain the difference with the term signaling pathways. Why did you decide to look for biological pathways?

Figure 4.

It is confusing. Both graphs goes from left to right even when showing -log(p-values)<0.

I suggest to put results in un graph only. It is easier to understand that the right part shows upregulated pathways (>0) and the left part shows downregulated ones (<0).

Also I would prefer to see these results expressed in z-score (X axis) like mentioned in materials&methods (line 185). Set the threshold line at 2 'Z‐score values ≥ 2 or <-2 were considered as significantly activated or inhibited'.

7. PLOS authors have the option to publish the peer review history of their article (what does this mean? ). If published, this will include your full peer review and any attached files.

**Do you want your identity to be public for this peer review?** For information about this choice, including consent withdrawal, please see our Privacy Policy .

Reviewer #1: No

Reviewer #2: No

---

## [Decision Letter · Decision Letter 2]

DIFFERENCES IN THE PROTEOMIC PROFILES OF THE EUTOPIC ENDOMETRIUM IN PATIENTS WITH INTERNAL AND EXTERNAL ADENOMYOSIS

PGPH-D-24-02496R2

Dear Miss Valdes-Bango Curell,

We are pleased to inform you that your manuscript 'DIFFERENCES IN THE PROTEOMIC PROFILES OF THE EUTOPIC ENDOMETRIUM IN PATIENTS WITH INTERNAL AND EXTERNAL ADENOMYOSIS' has been provisionally accepted for publication in PLOS Global Public Health.

Best regards,

Julia Robinson

Executive Editor

Reviewer Comments (if any, and for reference):

Reviewer's Responses to Questions

**Comments to the Author**

1. If the authors have adequately addressed your comments raised in a previous round of review and you feel that this manuscript is now acceptable for publication, you may indicate that here to bypass the “Comments to the Author” section, enter your conflict of interest statement in the “Confidential to Editor” section, and submit your "Accept" recommendation.

Reviewer #1: All comments have been addressed

Reviewer #2: All comments have been addressed

2. Does this manuscript meet PLOS Global Public Health’s publication criteria ? Is the manuscript technically sound, and do the data support the conclusions? The manuscript must describe methodologically and ethically rigorous research with conclusions that are appropriately drawn based on the data presented.

Reviewer #1: Yes

Reviewer #2: Yes

3. Has the statistical analysis been performed appropriately and rigorously?

Reviewer #1: Yes

Reviewer #2: Yes

4. Have the authors made all data underlying the findings in their manuscript fully available (please refer to the Data Availability Statement at the start of the manuscript PDF file)?

Reviewer #1: Yes

Reviewer #2: Yes

5. Is the manuscript presented in an intelligible fashion and written in standard English?

Reviewer #1: Yes

Reviewer #2: Yes

6. Review Comments to the Author

Reviewer #1: All questions have been addressed. I have no further questions regarding the current version of the manuscript.

Reviewer #2: The quality of the graphs is very low, making them appear blurred. However, I was still able to understand them thanks to the authors’ explanatory comments they provided and the detailed figure legends.

Finally, the authors have addressed all the comments raised.

7. PLOS authors have the option to publish the peer review history of their article (what does this mean? ). If published, this will include your full peer review and any attached files.

**Do you want your identity to be public for this peer review?** For information about this choice, including consent withdrawal, please see our Privacy Policy .

Reviewer #1: No

Reviewer #2: No
